

# The longitudinal associations between bone mineral density and appendicular skeletal muscle mass in Chinese community-dwelling middle aged and elderly men

Xuejuan Xu[1,2,3], Nuo Xu[4], Ying Wang[5], Jinsong Chen[3], Lushi Chen[6], Shengjian Zhang[6], Jingxian Chen[7], Hongwen Deng[8,9], Xiaojun Luan[3] and Jie Shen[1,2]

[1] Department of Endocrinology, Southern Medical University, Guangzhou, Guangdong, China
[2] Department of Endocrinology, The Third Affiliated Hospital of Southern Medical University, Guangzhou, Guangdong, China
[3] Department of Endocrinology, The First People's Hospital of Foshan, Foshan, Guangdong, China
[4] Department of Endocrinology, Affiliated Hospital of Jining Medical University, Jining Medical University, Jining, Shandong, China
[5] Department of Nuclear Medicine, The First People's Hospital of Foshan, Foshan, Guangdong, China
[6] Department of Health Care, The First People's Hospital of Foshan, Foshan, Guangdong, China
[7] Department of Hematology, The Eighth Affiliated Hospital, Sun Yat-sen University, Shenzhen, Guangdong, China
[8] Center of Genomics and Bioinformatics, Tulane University, New Orleans, LA, United States of America
[9] Department of Biostatistics and Bioinformatics, School of Public Health and Tropical Medicine, Tulane University, New Orleans, LA, United States of America

Corresponding authors
Xiaojun Luan, drluan@163.com
Jie Shen, sjiesy@smu.edu.cn

## ABSTRACT

**Background**. The present study aimed to investigate longitudinal associations between bone mineral densities (BMDs) and appendicular skeletal muscle (ASM) mass in different regions of the body using three different indicators, in Chinese community-dwelling middle-aged and elderly men.

**Methods**. A total of 1,343 men aged ≥ 40 years from a Chinese community were assessed at baseline (2014–2016), one-year follow-up (2016–2017; $n = 648$), two-year follow-up (2017–2018; $n = 407$), and three-year follow up (2018–2019; $n = 208$). At all the four time-points, measurements included ASM mass and BMDs for all regions of the body using dual-energy X-ray absorptiometry. A questionnaire was completed by patients and biochemical markers were assessed. We applied three different indicators to define ASM mass or lean mass respectively, including the appendicular skeletal muscle index (ASM adjusted by height, ASMI, according to the Asian Working Group for Sarcopenia), skeletal muscle index (ASM adjusted by weight, SMI, according to the International Working Group on Sarcopenia), and the appendicular skeletal muscle/body mass index (ratio of ASM and Body mass index (BMI), ASM/BMI, according to the Foundation for the National Institutes of Health). After adjusting for potential confounders, the generalized additive mixed model (GAMM) was used to analyze the trend in ASM mass over time, and to test the association between ASM mass and regional and whole-body BMDs.

**Results**. The incidence of low lean mass was 8.2% defined by ASMI, 16.3% defined by SMI, and 8.3% defined by ASM/BMI. There was a linear relationship between BMDs and ASM mass, and ASMI, ASM/BMI, and SMI gradually decreased with time. After adjusting for covariances, GAMM analysis determined longitudinal associations between BMDs and ASM mass by three indicators respectively: the skull BMD was negatively associated with ASM mass. For each unit increase in skull BMD, ASMI decreased by 0.28 kg/m$^2$ (95% confidence interval (CI) [−0.39 to −0.16]), ASM/BMI decreased by 0.02 m$^2$ (95% CI [−0.03 to −0.00]), and SMI decreased by 0.01% (95% CI[−0.01 to −0.00]). The remaining parameters (including whole-body mean BMD, thoracic spinal BMD, lumbar spinal BMD, hip BMD, femoral neck BMD, pelvic BMD, left arm BMD, right arm BMD, left leg BMD, right leg BMD) were positively correlated with ASM mass. The ASMI increased by 3.07 kg/m$^2$ for each unit increase in the femoral neck BMD (95% CI [2.31–3.84]). The ASM/BMI increased by 0.22 m$^2$ for each unit increase in the left arm BMD (95% CI [0.12–0.33]), and the SMI increased by 0.05% per unit increase in the left arm BMD (95% CI [0.02–0.08]).

**Conclusions**. Compared to ASMI and ASM/BMI, SMI was more sensitive to screen for the low lean mass. Skull BMD was negatively associated with ASM mass, while BMDs throughout the rest of the body were positively correlated with ASM mass among the middle-aged and elderly Chinese men.

# INTRODUCTION

The aging of a population leads to an upsurge of age-related diseases, in which sarcopenia and osteoporosis are attracting increasing attention (*Binkley, Krueger & Buehring, 2013*; *Kim et al., 2017a*). It is thought that sarcopenia is associated with osteoporosis as an activity disorder syndrome (dysmotility syndrome) (*Marty et al., 2017*), which consumes considerable health and social costs (*Pinedo-Villanueva et al., 2019*; *Robinson et al., 2018*). The muscles of individuals younger than 25 years of age are in the ascending stage and appendicular skeletal muscle (ASM) remains stable in 25th–40th years of age. After 40 year old, ASM mass generally declines (*Morley, Anker & Von Haehling, 2014*).

The interrelationships of sarcopenia and osteoporosis are complex and multifactorial (*Karasik & Kiel, 2010*). Many studies had detailed the effects of skeletal muscle on bone tissue (*Colaianni et al., 2016*; *Pedersen & Febbraio, 2012*). Recently, bone cells have been considered as endocrine cells and can transmit signals to distant organs including muscles (*Brotto & Johnson, 2014*; *Brun et al., 2017*). Additionally, osteoblasts of the craniofacial bones, which are different from other bone tissues of the body, are derived from neural crest cells differentiated from the neuroectoderm and are regulated by different mechanism (*Gilbert, 2000*; *Vatsa et al., 2008*; *Wu et al., 2018*). Therefore, the interrelationship of skull and ASM may be different from that of the bone tissues throughout the rest of the body and ASM through the circulatory system. However, these hypothesis require

further experimental confirmation. Understanding the apparent endocrine crosstalk and biochemical coupling between these two intimately associated tissues (bone and muscle) is important for identifying potential new therapies for the relevant diseases, especially for when they co-exist.

Some studies reported that bone mineral densities (BMDs) and muscle mass are positively associated (*Bering et al., 2018*; *Rodriguez-Reyes et al., 2019*). In 2016, *He et al., (2016)* had recruited 17,891 African Americans, Chinese and Caucasians, and found that ASM mass was positively correlated with the whole-body mean BMD, femoral neck BMD, tibia BMD, and lumbar spine BMD. However, several studies reported that sarcopenia is not associated with BMD. *Coin et al. (2008)* reported that in 136 elderly men ASM mass was positively associated with hip and femoral neck BMDs, but after adjustment by body mass index (BMI), the association was not significant. *Walsh, Hunter & Livingstone (2006)* investigated 130 premenopausal and 82 postmenopausal African American women living in the United States and found that a decreased in appendicular skeletal muscle index (ASM adjusted by height, ASMI), was not associated with low hip BMD. However, a small population and only measuring BMD of hip or femur can lead to a failure to reflect the BMDs throughout the rest of the body in studying their relationship with muscle (*Coin et al., 2008*; *Genaro et al., 2010*; *Walsh, Hunter & Livingstone, 2006*), because bone cells of different parts of the body have different roles and functions, such as fibula and skull (*Vatsa et al., 2008*). Additionally, the association of BMD of hip or femur and lean mass is largely due to mechanical coupling (*Cianferotti & Brandi, 2014*). These contradictory findings might be related to the differences of study population, body part, measuring method and indicator.

Most clinical research were cross-sectional studies, and few prospective research has revealed the longitudinal associations of ASM mass and regional BMDs. Cross-sectional studies may have had limited statistical power to find evidence of longitudinal associations given the sample size (*Martinez et al., 2017*). *Genaro et al., (2010)* had investigated 65 postmenopausal osteoporotic women to find that lean mass is positively correlated with BMDs of femoral neck and total femur. One prospective 10-year study investigated 104 normal white postmenopausal women and demonstrated that only fat mass is associated with total body, femur and spine BMDs, but not lean mass (*Wu et al., 2002*). No prospective study had enrolled big sample size to study the associations between the BMDs of different regions of the body and changes in ASM mass.

According to the European Working Group on Sarcopenia in Older People (EWGSOP) and the Asian Working Group for Sarcopenia (AWGS) (*Chen et al., 2014*; *Cruz-Jentoft et al., 2010*), there are three stages of sarcopenia that reflect the severity of the condition, including 'presarcopenia', 'sarcopenia' and 'severe sarcopenia'. The 'presarcopenia' stage is characterized by low lean mass without poor muscle strength or physical performance. Currently, there are three indicators for the measurement of ASM mass to diagnose sarcopenia, including the ASMI, skeletal muscle index (ASM adjusted by weight, SMI), and the appendicular skeletal muscle/body mass index (ratio of ASM and Body mass index (BMI), ASM/BMI). Different consensuses recommend different indicators for the measurement of ASM mass and cut-off value for low lean mass (*Chen et al., 2016*;

*Cruz-Jentoft et al., 2010*; *Muscaritoli et al., 2010*; *Studenski et al., 2014*). Due to differences in ethnicity, genetic background, and body size, the EWGSOP and International Working Group on Sarcopenia (IWGS) criteria might not be suitable to Asians. The EWGSOP recommends using the ASMI. The AWGS proposed a cut-off value for the definition of low lean mass using the ASMI in Asian populations. Using dual-energy X-ray absorptiometry (DXA), the threshold was 7.0 kg/m2 for men and 5.4 kg/m2 for women (*Chen et al., 2014*). The Foundation for the National Institutes of Health (FNIH) applied ASM/BMI, and the IWGS recommends using the SMI. The prevalence and clinical implications of sarcopenia varies greatly depending on the indicators for the measurement of ASM mass (*Kim, Jang & Lim, 2016*; *Scott et al., 2017*). Studies in South Korea and Taiwan had found that applying the SMI can more accurately detect sarcopenia than using the ASMI (*Hairi et al., 2010*; *Kim, Jang & Lim, 2016*). Most previous studies used one indicator to define ASM mass, which makes it impossible to compare the data across studies (*Chung et al., 2016*; *Kim et al., 2014*). Comparisons of the prevalence of low lean mass using the three indicators (ASMI, SMI or ASM/BMI) respectively have not been reported in China.

To date, the longitudinal associations between the BMDs of different regions of the body and changes in ASM mass defined by the three different indicators respectively has not been thoroughly studied. In the current study, we assessed such associations using the three indicators respectively and a 3 years of follow-up for regional and whole-body BMDs in Chinese community-dwelling middle aged and elderly men.

## MATERIALS & METHODS

### Study design and population

A prospective study was conducted on 3,179 males with detailed data on body composition and BMDs for all regions of the body. The study was performed in the Health Care Department in the Foshan First People's Hospital from January 2014 to September 2016, and all patients were from Chinese community. According to patient histories and examination results, the following subjects were excluded: patients younger than 40 years of age, those with complete walking incapacity, viral or autoimmune hepatitis, other chronic liver diseases, alcohol-addicted (>210 g alcohol per week), severe heart or kidney dysfunction, severe dementia (mini-mental state examination <18 points), use of steroid hormones or immunosuppressive agents, autoimmune diseases, use of weight-loss drugs, pathological obesity, uncontrollable diabetes, hypothyroidism or other endocrine and metabolic diseases, or a diagnosis with malignant tumors in the past five years and acute disease stages. The study design was registered and approved by the Ethics Committee of the Foshan First People's Hospital. All the patients and their families understood and agreed with the purpose of the study, and signed the informed consent for the study. Body composition and BMD were repeated over the following 3 years. Data were collected from January 2014 to September 2018.

This study was approved by the hospital ethics committee with the informed consent of the children's family, which was conducted ethically in accordance with the World Medical Association Declaration of Helsinki.

## Measurement of anthropometric indicators

Subjects fasted for 8 h and emptied their bowels. BMI was calculated as: BMI $(kg/m^2)$ = body mass $(kg)/[height (m)^2]$. Blood pressure was measured after 20 min of sit-down.

## Blood sample measurements

Blood samples were obtained from the forearm veins of all subjects after fasting for longer than 8 h. The content of plasma homocysteine was determined by high performance liquid chromatography. Serum albumin was analyzed by turbidimetry. We used an automatic biochemical analyzer (OLYMPUS AU5400; Olympus, Shinjuku, Tokyo, Japan) to measure biochemical indices: (1) blood lipid profile which included total cholesterol (Cholesterol), triglycerides, high density lipoprotein-cholesterol (HDL-C), low density lipoprotein-cholesterol, lipoprotein a, Apolipoprotein A1, Apolipoprotein B; (2) full blood count which mainly included platelet width distribution, platelet count, variation of red blood cell distribution, standard deviation of red blood cell distribution; (3) thyroid function which included thyroid stimulating hormone, free triiodothyronine, free thyroxine (FT4); (4) renal function which included urea nitrogen, creatinine (Cr), uric acid (UA); (5) liver function which included aspartate aminotransferase (AST), alanine aminotransferase (ALT), gamma-glutamyl transpeptidase (GGT), total bilirubin, direct bilirubinm, indirect bilirubin, albumin, globulin, total protein ; (6) blood glucose which included fasting blood glucose (FBG), postprandial blood glucose (PBG), glycosylated hemoglobin (HbA1c); (7) trace minerals which included calcium, phosphorus; (8) Homocysteine.

## Body composition and BMD

Body composition and BMD $(g/cm^2)$ were measured using a DXA scanner (HOLOGIC, model: discovery A). Such measurements were obtained for the whole-body mean BMD, hip, femoral neck, lumbar spine, left upper arm, right upper arm, left leg, right leg, left rib, right rib, thoracic vertebra, pelvis, and skull. Densitometers were calibrated daily using the method provided by the manufacturer to ensure the accuracy of DXA measurements. The coefficient of variation (percentage) measured repeatedly in 30 adults was in accordance with the accurate criteria of the lumbar spine (<1.9%), femoral neck (2.5%), and total femur (<1.8%).

Body composition included muscle mass, weight after fat removal (FFM), and fat mass (FM). The coefficients of variation for whole body fat (FM) and FFM were 0.89% and 0.48%, respectively. ASM mass was defined as the skeletal muscle mass of the extremities.

## Definitions of low lean mass

We used three indicators to diagnose low lean mass in all male participants respectively: (1) an ASMI <7.0 $kg/m^2$. This approach for defining low lean mass was based on the new Asian consensus definition of the AWGS (*Chen et al., 2016*); (2) a SMI <29.9% (*Kim, Cho & Park, 2015*; *Ryu et al., 2013*). This approach for defining low lean mass was based on some Asian studies carried out in Korea; (3) an ASM/BMI <0.789 $m^2$. There are no recommended cut-off value of ASM/BMI for Asian patients. Thus, we use the cut-off value recommended by FNIH (*Studenski et al., 2014*).The control groups are defined as: (1) ASMI ≥7.0 $kg/m^2$; (2) SMI ≥29.9%; or (3) ASM/BMI ≥0.789 $m^2$.

## Sociodemographic and lifestyle status

Comprehensive interviewer-assisted questionnaires were conducted regarding demographic, occupational, and lifestyle information, alcohol consumption, smoking histories, diet, exercise, and mental health. According to the results of the questionnaire, the self-reported smoking, drinking, exercise and mental status were assessed. Subjects who never smoked or quit smoking at least 5 years prior to study were defined as nonsmokers. Subjects were considered to consume alcohol if they drank any volume of alcohol at least once per week. Regular exercise was defined by any type of exercise at least once per week.

## Statistical analyses

All analyses were performed using Empower (R) (http://www.empowerstats.com, X&Y solutions, Inc., Boston, MA) and R (http://www.R-project.org). All statistical analyses were two-tailed, and a $P < 0.05$ was considered statistically significant. The means and standard deviations were calculated for anthropometric measures. Differences in basic characteristics were compared using analysis of variance for continuous variables. The Pearson's chi-squared test ($\chi^2$) was used to compare differences for categorical variables. All $P$ values were corrected for multiple hypothesis testing using the false-discovery rate (Benjamini–Hochberg method), with a significance threshold of 5%. And the corrected $P$ value meant Q valuewhich could better avoid false positive error.

We applied a two-piecewise linear regression model to examine the threshold effect of BMD on ASM mass (defined by ASMI, SMI, and ASM/BMI) using a smoothing function (*Yu, Cao & Yu, 2013*). The threshold level was determined by trial and error, including the selection of turning points along a pre-defined interval and choosing the turning point that gave the maximum model likelihood, after adjusting for age, body weight, ALT, GGT, FT4, triglyceride, HDL-C, Cr, and UA.

The generalized additive mixed model (GAMM) was used to analyze trends in ASM mass over time. Models were initially unadjusted and then adjusted for age, weight, and HbA1c.

To study the longitudinal association between changes in BMD of different regions of the body and changes in the ASM mass, we used the GAMM to analyze (*Canfield et al., 2003*). GAMM takes into account the time-varying nature of both the outcome and the exposure over multiple time-points and provides an estimated population average model using all longitudinal data. With GAMM analysis, the association between two longitudinally measured variables can be studied using all longitudinal data simultaneously, and adjusting for within-person correlations caused by repeated measurements of each participant using robust estimations of the variances of the regression coefficients. GAMM is also robust with regard to data missing at random. Models were initially unadjusted and then adjusted for age, weight, HbA1c, HDL-C, Cr, ALT, FT4, diastolic blood pressure, smoking, drinking, and exercise.

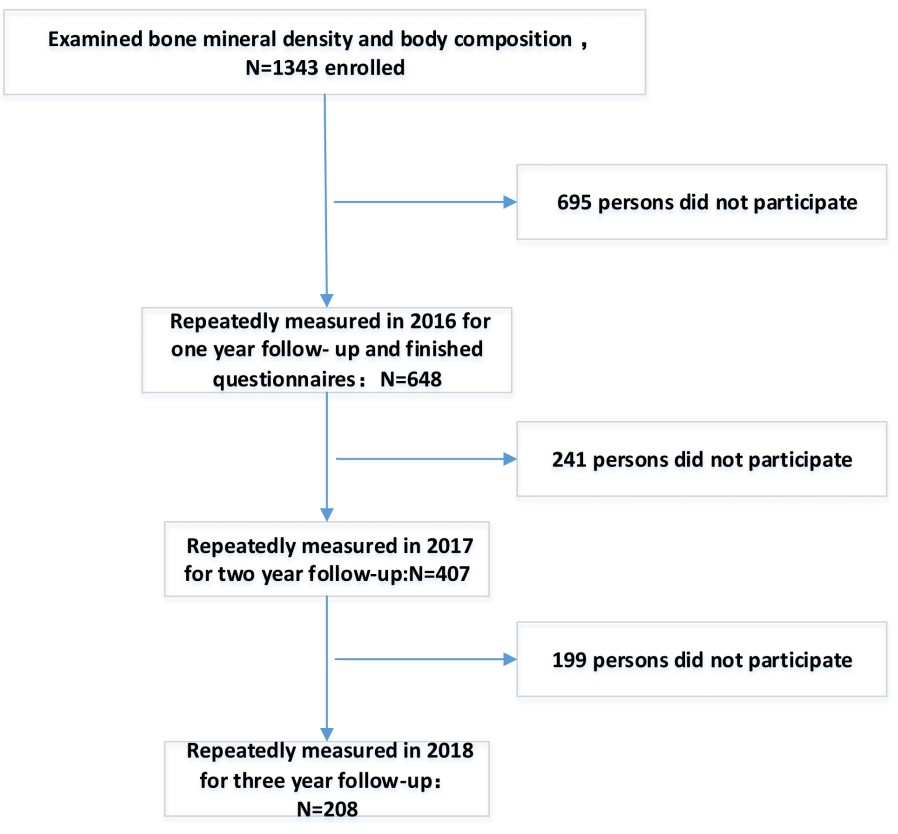

Figure 1    Flowchart of participation in this study.

## RESULTS

### Descriptive statistics

After excluding participants who did not attend baseline, a total of 1,343 participants were included in the analysis. Figure 1 is a participation flowchart of the 1,343 Chinese males included in this study. Of the 1,343 participants with complete baseline data, 648 (48.25%) completed the one-year follow-up, 407 (30.3%) completed the two-year follow-up, and 208 (15.48%) completed the three-year follow-up. BMD and muscle mass were examined in all regions of the body each year. Reasons for the study's loss of follow-up included unreachable, refusal of follow-up or death.

Table 1 presents the basic characteristics, anthropometric measurements, and regional and total body muscles (including head, left arm, right arm, trunk, left leg, right leg, whole body), and BMDs of the study population, which was stratified into two groups based on the ASMI. ASMI <7.0 kg/m2 was used to define low-lean- mass group, and the control group was defined ASMI ≥7.0 kg/m2. The incidence of low lean mass was 8.2% in individuals older than 40 years of age. Compared to control group, the low-lean- mass group was shorter (167.95 ± 6.46 cm and 169.92 ± 5.72, $Q$-value = 0.002) and lighter (60.83 ± 7.56 kg and 73.63 ± 8.79 kg, $Q$-value = 0.002). Compared with the control group, the muscle mass of regional and total body of the low-lean- mass group were lower.

**Table 1  Data of low-lean-mass and control group (mean ± s) and diagnosis according to ASMI <7.0.**

| | Low-lean-mass group ($n = 121$) | Control group ($n = 1222$) | *P* value | *Q* value |
|---|---|---|---|---|
| **Anthropometric measurement** | | | | |
| Age (years) | 59.73 ± 12.81 | 54.40 ± 7.46 | <0.001 | 0.002 |
| Weight (kg) | 60.83 ± 7.56 | 73.63 ± 8.79 | <0.001 | 0.002 |
| Height (cm) | 167.95 ± 6.46 | 169.92 ± 5.72 | <0.001 | 0.002 |
| BMI (kg/m$^2$) | 21.56 ± 2.41 | 25.49 ± 2.62 | <0.001 | 0.002 |
| Systolic blood pressure (mmHg) | 127.51 ± 17.63 | 125.87 ± 15.50 | 0.377 | 0.537 |
| Diastolic blood pressure (mmHg) | 75.32 ± 10.11 | 79.38 ± 10.72 | <0.001 | 0.002 |
| Heart rate (times/min) | 74.83 ± 9.99 | 72.56 ± 10.42 | 0.049 | 0.096 |
| **Body composition** | | | | |
| HEAD_LEAN (g) | 3771.97 ± 330.24 | 4048.45 ± 324.55 | <0.001 | 0.002 |
| LARM_LEAN (g) | 2667.17 ± 335.92 | 3439.74 ± 463.65 | <0.001 | 0.002 |
| RARM_LEAN (g) | 2925.54 ± 388.82 | 3775.33 ± 482.83 | <0.001 | 0.002 |
| TRUNK_LEAN (g) | 22341.13 ± 2647.00 | 26979.66 ± 3327.26 | <0.001 | 0.002 |
| L_LEG_LEAN (g) | 7002.52 ± 774.36 | 8828.57 ± 1039.66 | <0.001 | 0.002 |
| R_LEG_LEAN (g) | 7011.08 ± 961.50 | 8934.63 ± 1052.54 | <0.001 | 0.002 |
| WBTOT_LEAN (g) | 45719.41 ± 4599.27 | 56006.38 ± 5956.41 | <0.001 | 0.002 |
| **Biochemical metabolic markers** | | | | |
| ALT (IU/L) | 21.03 ± 10.29 | 26.64 ± 16.35 | <0.001 | 0.002 |
| AST (IU/L) | 19.99 ± 4.69 | 22.20 ± 10.06 | 0.024 | 0.049 |
| ALP (IU/L) | 63.05 ± 13.44 | 61.28 ± 15.06 | 0.194 | 0.329 |
| GGT (IU/L) | 33.80 ± 25.03 | 41.52 ± 33.50 | <0.001 | 0.002 |
| Total protein (g/L) | 74.00 ± 3.77 | 74.48 ± 3.91 | 0.258 | 0.402 |
| Albumin (g/L) | 43.17 ± 2.49 | 43.52 ± 2.53 | 0.219 | 0.349 |
| Globulin (g/L) | 30.83 ± 3.12 | 30.81 ± 3.53 | 0.843 | 0.913 |
| Total bilirubin (µmol/L) | 15.13 ± 5.67 | 15.19 ± 5.67 | 0.583 | 0.7 |
| Direct bilirubin (µmol/L) | 5.18 ± 2.79 | 5.05 ± 2.87 | 0.798 | 0.877 |
| Indirect bilirubin (µmol/L) | 9.94 ± 4.46 | 10.14 ± 4.30 | 0.317 | 0.476 |
| Fasting blood glucose (mmol/L) | 5.25 ± 1.30 | 5.30 ± 1.24 | 0.356 | 0.524 |
| Postprandial blood glucose (mmol/L) | 6.17 ± 2.65 | 6.49 ± 2.56 | 0.152 | 0.265 |
| HbA1c (%) | 5.86 ± 1.47 | 5.77 ± 0.72 | 0.307 | 0.47 |
| Total cholesterol (mmol/L) | 5.16 ± 0.95 | 5.12 ± 0.96 | 0.448 | 0.573 |
| Triglycerides (mmol/L) | 1.58 ± 0.95 | 1.95 ± 1.66 | 0.010 | 0.022 |
| Apolipoprotein A1 (g/L) | 1.56 ± 0.31 | 1.43 ± 0.30 | 0.217 | 0.349 |
| Apolipoprotein B (g/L) | 1.06 ± 0.24 | 1.08 ± 0.26 | 0.583 | 0.7 |
| HDL-C (mmol/L) | 1.30 ± 0.25 | 1.21 ± 0.26 | <0.001 | 0.002 |
| LDL-C (mmol/L) | 2.94 ± 0.73 | 2.97 ± 0.79 | 0.757 | 0.868 |
| Lpa (mg/dL) | 227.06 ± 195.22 | 212.39 ± 289.86 | 0.136 | 0.253 |
| TSH (µTU/mL) | 1.84 ± 1.08 | 1.76 ± 1.62 | 0.123 | 0.234 |
| FT3 (pmol/L) | 5.01 ± 0.84 | 4.99 ± 0.60 | 0.932 | 0.957 |
| FT4 (pmol/L) | 17.67 ± 3.08 | 16.97 ± 2.38 | 0.020 | 0.04 |

**Table 1** (*continued*)

|  | Low-lean-mass group (*n* = 121) | Control group (*n* = 1222) | *P* value | *Q* value |
|---|---|---|---|---|
| Creatinine (μmol/L) | 78.16 ± 12.50 | 81.63 ± 12.83 | 0.003 | 0.007 |
| Urea nitrogen (μmol/L) | 5.44 ± 1.47 | 5.37 ± 1.19 | 0.897 | 0.945 |
| Uric acid (μmol/L) | 393.79 ± 85.32 | 438.58 ± 90.55 | <0.001 | 0.002 |
| Phosphorus (mmol/L) | 4.65 ± 1.56 | 4.68 ± 1.56 | 0.977 | 0.977 |
| Calcium (mmol/L) | 2.30 ± 0.09 | 2.30 ± 0.10 | 0.797 | 0.877 |
| Homocysteine | 12.90 ± 3.35 | 12.71 ± 4.35 | 0.401 | 0.537 |
| White blood cell count (10$^9$) | 6.31 ± 1.53 | 6.35 ± 1.62 | 0.945 | 0.957 |
| Platelet distribution width (fl) | 14.13 ± 2.42 | 14.46 ± 2.17 | 0.145 | 0.263 |
| Standard deviation of red blood cell distribution | 41.69 ± 2.99 | 41.71 ± 2.69 | 0.404 | 0.537 |
| Variation coefficient of red blood cell distribution | 0.13 ± 0.01 | 0.13 ± 0.01 | 0.787 | 0.877 |
| **Demographic and lifestyle status** | | | | |
| Alcohol drinking (%) | 12% | 11% | 0.931 | 0.957 |
| Smoking (current smoker) (%) | 17% | 15.8% | 0.567 | 0.7 |
| High fat diet (%) | 35.1% | 30.3% | 0.423 | 0.55 |
| Coffee or tea intake (%) | 14.2% | 15.7% | 0.731 | 0.851 |
| Regular exercise (%) | 56.3% | 58.9% | 0.153 | 0.265 |
| Family income (ten thousand RMB/per year) | 10 ± 1 | 9 ± 3 | 0.406 | 0.537 |
| Mental stress (%) | 6.4% | 7% | 0.216 | 0.349 |
| **Bone density (g/cm$^2$)** | | | | |
| WBTOT_BMD | 1.07 ± 0.09 | 1.12 ± 0.09 | <0.001 | 0.002 |
| HEAD_BMD | 2.14 ± 0.34 | 2.15 ± 0.31 | 0.894 | 0.945 |
| LARM_BMD | 0.74 ± 0.06 | 0.78 ± 0.06 | <0.001 | 0.002 |
| RARM_BMD | 0.77 ± 0.06 | 0.81 ± 0.06 | <0.001 | 0.002 |
| LRIB_BMD | 0.56 ± 0.06 | 0.60 ± 0.08 | <0.001 | 0.002 |
| RRIB_BMD | 0.58 ± 0.08 | 0.60 ± 0.07 | <0.001 | 0.002 |
| T_S_BMD | 0.81 ± 0.11 | 0.88 ± 0.11 | <0.001 | 0.002 |
| L_S_BMD | 0.94 ± 0.15 | 1.00 ± 0.14 | <0.001 | 0.002 |
| PELV_BMD | 1.11 ± 0.15 | 1.24 ± 0.17 | <0.001 | 0.002 |
| LLEG_BMD | 1.09 ± 0.10 | 1.17 ± 0.10 | <0.001 | 0.002 |
| RLEG_BMD | 1.09 ± 0.09 | 1.17 ± 0.10 | <0.001 | 0.002 |
| HIP_BMD | 0.89 ± 0.13 | 0.96 ± 0.12 | <0.001 | 0.002 |
| HIPNECK_BMD | 0.73 ± 0.12 | 0.80 ± 0.12 | <0.001 | 0.002 |

**Notes.**

Data are presented as mean ± SE or number.

ASMI, appendicular skeletal muscle index; HEAD_LEAN, lean mass of head; LARM_LEAN, lean mass of left arm; RARM_LEAN, lean mass of right arm; TRUNK_LEAN, lean mass of trunk; L_LEG_LEAN, lean mass of left leg; R_LEG_LEAN, lean mass of right leg; WBTOT_LEAN, lean mass of whole body; HbA1c, glycosylated hemoglobin; AST, aspartate aminotransferase; ALT, alanine aminotransferase; HDL-C, high density lipoprotein; LDL-C, low density lipoprotein; TSH, thyroid stimulating hormone; FT3, free triiodothyronine; FT4, free thyroxine; WBTOT_BMD, mean whole-body BMD; HEAD_BMD, skull BMD; LRIB_BMD, left rib BMD; RRIB_BMD, right rib BMD; T_S_BMD, thoracic spinal BMD; L_S_BMD, lumbar spinal BMD; PELV_BMD, pelvic BMD; HTOT_BMD, hip BMD; NECK_BMD, femoral neck BMD; LLEG_BMD, left leg BMD; RLEG_BMD, right leg BMD; LARM_BMD, left arm BMD; RARM_BMD, right arm BM.

The BMDs of regional and total body in the low-lean- mass group was lower, with the exception of that of the skull. Of the biochemical metabolic markers, ALT, AST, GGT, triglyceride, HDL-C, UA, FT4, and Cr were different between control and low-lean- mass groups (*Q*-value < 0.05). There was no significant difference in serum albumin levels between the two groups, which represent nutritional status. The two groups had similar blood glucose levels, including FBG, PBG, and glycosylated hemoglobin. The inflammatory indicators were also similar, such as the total number of leukocytes, platelet distribution width, coefficient of variation of erythrocyte distribution, standard deviation of erythrocyte distribution, and homocysteine levels.

Table S1 shows the data for the study population, which was stratified into two groups according to SMI. SMI <29.9% was used to defined low-lean- mass group, and the control group was defined as SMI ≥29.9%. The incidence of low lean mass was 16.3%. Compared to the control group, the muscle mass of the head and trunk of individuals in the low-lean-mass group were larger, while the other parts were lower (including left arm, right arm, left leg, right leg and whole body), and the bone density of each part (including whole body, head, left arm. right arm. trunk, left leg, and right leg) was lower.

Table S2 shows the data for the study population which was stratified into two groups based on ASM/BMI. ASM/BMI <0.789 $m^2$ was used to defined low-lean- mass group, and the control group was defined as ASM/BMI ≥0.789 $m^2$. The incidence of low lean mass was 8.3%. Regional and total body muscles in the low-lean- mass group was lower than that of the control group, while the bone densities of regional and total body were lower; thus, indicating that the three indicators (ASMI, SMI, ASM/BMI) cannot be used interchangeably (*Kim et al., 2017b*).

## Threshold effect analysis of relationship between BMDs and ASM mass

After adjusting for diastolic blood pressure, age, body weight, ALT, glycosylated hemoglobin, triglyceride, HDL-C, UA, FT4, and Cr, smooth curve fitting analysis evaluated relationships between BMDs and the three indicators (ASMI, SMI, ASM/BMI) to determine if there was a threshold effect. Adjusted smoothed plots suggest a linear relationship between BMDs and ASM mass (Figs. 2A–2M).

## Changes of ASM mass over time

GAMM was used to analyze trends in muscle mass over time. The ASMI, ASM/BMI, and SMI were plotted on the ordinate, and the number of days was plotted on the abscissa (total 1,200 days). The number of days was calculated based on the date of examination. It was found that the ASMI, ASM/BMI, and SMI gradually decreased with time, regardless of whether adjustments for age, weight, and HbA1c were performed (Figs. 3A–3C).

## Repeated measurements to analyze the associations between BMDs and ASM mass

GAMM analysis determined average longitudinal population associations between BMDs and ASM mass, after adjusting for age, weight, HbA1c, HDL-C, Cr, ALT, FT4, diastolic blood pressure, smoking, drinking and exercise over four time points.

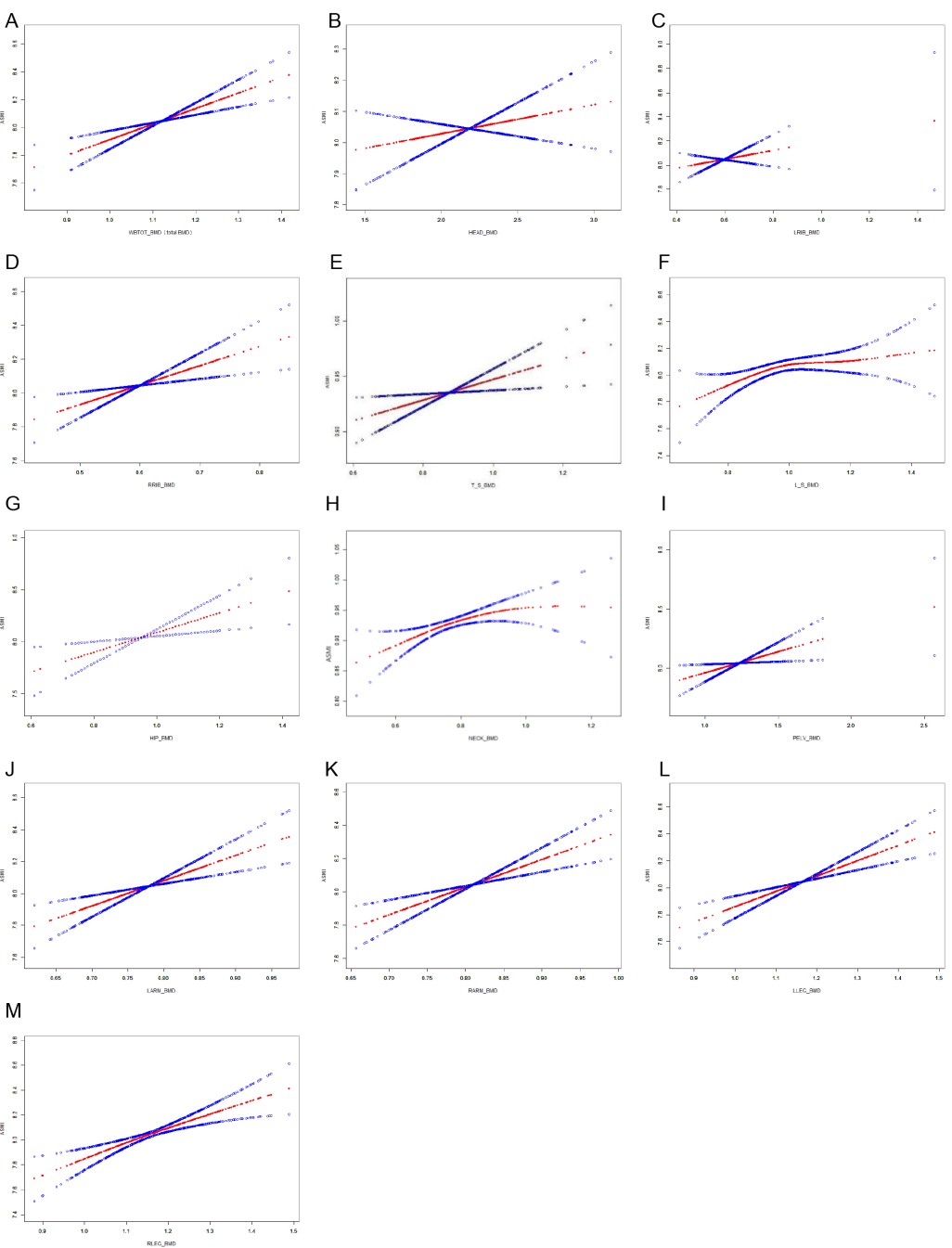

**Figure 2** **ASMI and BMDs relationships.** (A) ASMI and mean whole-body BMD. (B) ASMI and skull BMD. (C) ASMI and Left rib BMD. (D) ASMI and right rib BMD. (E) ASMI and thoracic spinal BMD. (F) ASMI and Lumbar spinal BMD. (G) ASMI and hip BMD. (H) ASMI and Femoral neck BMD. (I) ASMI and pelvic BMD. (J) ASMI and Left arm BMD. (K) ASMI and right arm BMD. (L) ASMI and Left leg BMD. (M) ASMI and right leg BMD.

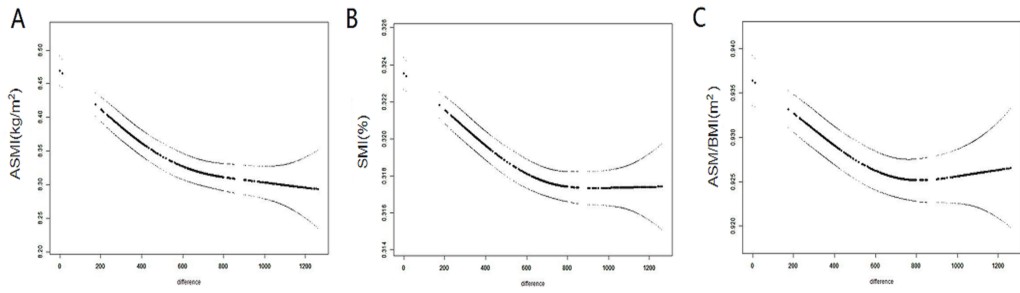

**Figure 3** **Changes of ASM over time.** (A) Trends of ASMI over time. (B) Trends of SMI over time. (C) Trends of ASM/BMI over time.

The ASMI, ASM/BMI, and SMI were treated as the dependent variables (result variable), and after adjustment, the BMD of skulls were negatively correlated with the ASMI, ASM/BMI and SMI. For each decreasing unit of skull BMD, the ASMI increased by $0.28^2$ [95% confidence interval (CI) [−0.39, −0.16]; $P < 0.001$] (Table 2), while the SMI increased by 0.01% (95% CI [−0.01, −0.00]; $P = 0.009$) (Table S3), and the ASM/BMI increased by 0.02 $m^2$ (95% CI [−0.03, −0.00]; $P = 0.008$) (Table S4). Conversely, the BMDs of the pelvis, hip and femur neck had positive correlations with the ASMI, but were not correlated with SMI and ASM/BMI. The BMDs of left and right ribs were not correlated with three indicators according to the sensitivity analysis (Tables S5–S7). The remaining parts (including whole body, thoracic spinal, lumbar spinal, left leg, right leg, left arm and right arm) were positively correlated with the three indicators (ASMI, SMI, ASM/BMI). The highest beta value for BMD at each site was that for every unit of increase in femoral neck BMD, the ASMI increased by 3.07 $m^2$ (95% CI [2.31–3.84]; $P < 0.001$). For each unit ($g/cm^2$) increase in the left arm BMD, the ASM/BMI increased by 0.22 $m^2$ (95% CI [0.12–0.33]; $P < 0.001$), while the SMI increased by 0.05% (95% CI [0.02–0.08]; $P < 0.001$).

## DISCUSSION

This prospective population-based study demonstrated that compared to ASMI and ASM/BMI, SMI was more sensitive to screen for the low lean mass, and skull BMD was negatively correlated with ASM mass defined by the ASMI, ASM/BMI or SMI. Conversely, the BMDs of the other regions of the body were positively correlated with the ASM mass, which have not been reported in prior studies.

This is the first study to measure BMD using three different indicators (ASMI, SMI and ASM/BMI) for four consecutive years and to observe BMD in all regions of the body. The incidence of low lean mass defined by SMI is nearly two times of that by ASMI or ASM/BMI, which revealed SMI was more sensitive to screen for the low lean mass. This is echoed by the previous reports (*Clynes et al.,2015*; *Kim, Jang & Lim, 2016*) (Clynes MA, Edwards MH, Buehring B, Dennison EM, Binkley N, Cooper C. 2015. Definitions of Sarcopenia: Associations with Previous Falls and Fracture in a Population Sample.Calcif Tissue Int. 97(5):445-52. DOI: 10.1007/s00223-015-0044-z) . The IWGS definition of

**Table 2 Associations between low lean mass changes according to ASMI and BMDs (n = 1,343).**

| Outcome:*ASMI | Unadjusted β coefficient (95% CI) | | | Adjusted* β coefficient (95% CI) | | |
|---|---|---|---|---|---|---|
| | β | (95% CI) | P | β | (95% CI) | P |
| WBTOT_BMD | 0.84 | (0.26, 1.42) | 0.005 | 0.74 | (0.17, 1.30) | 0.011 |
| HEAD_BMD | −0.3 | (−0.42, −0.19) | <0.001 | −0.28 | (−0.39, −0.16) | <0.001 |
| LRIB_BMD | −0.95 | (−1.44, −0.47) | <0.001 | −0.96 | (−1.44, −0.48) | <0.001 |
| RRIB_BMD | −0.41 | (−0.75, −0.06) | 0.021 | −0.36 | (−0.70, −0.02) | 0.04 |
| T_S_BMD | 0.98 | (0.63, 1.34) | <0.001 | 1.06 | (0.71, 1.40) | <0.001 |
| L_S_BMD | 0.4 | (0.07, 0.73) | 0.017 | 0.45 | (0.12, 0.77) | 0.007 |
| PELV_BMD | 1.35 | (1.04, 1.66) | <0.001 | 1.19 | (0.89, 1.50) | <0.001 |
| HTOT_BMD | 3.11 | (2.41, 3.81) | <0.001 | 3.06 | (2.38, 3.75) | <0.001 |
| NECK_BMD | 3.2 | (2.43, 3.97) | <0.001 | 3.07 | (2.31, 3.84) | <0.001 |
| LLEG_BMD | 2.07 | (1.58, 2.56) | <0.001 | 1.85 | (1.36, 2.33) | <0.001 |
| RLEG_BMD | 2.03 | (1.56, 2.51) | <0.001 | 1.84 | (1.37, 2.31) | <0.001 |
| LARM_BMD | 2.92 | (2.04, 3.80) | <0.001 | 2.56 | (1.69, 3.42) | <0.001 |
| RARM_BMD | 2.59 | (1.78, 3.40) | <0.001 | 2.21 | (1.40, 3.01) | <0.001 |

**Notes.**
*Adjusted for age, weight, HbA1c, HDL-C, creatinine, ALT, FT4, diastolic blood pressure, smoking, drinking and exercise.
ASMI, appendicular skeletal muscle index; CI, confidence interval; WBTOT_BMD, whole-body BMD; HEAD_BMD, skull BMD; LRIB_BMD, left rib BMD; RRIB_BMD, right rib BMD; T_S_BMD, thoracic spinal BMD; L_S_BMD, lumbar spinal BMD; PELV_BMD, pelvic BMD; HTOT_BMD, hip BMD; NECK_BMD, femoral neck BMD; LLEG_BMD, left leg BMD; RLEG_BMD, right leg BMD; LARM_BMD, left arm BMD; RARM_BMD, right arm BMD.

sarcopenia appears to be an effective means of identifying individuals at risk of prevalent adverse musculoskeletal events.

Most studies observed the relationship between muscle mass and the BMDs of the femoral neck, or lumbar spine, which, to a large extent, reflects the mechanical action between these closely related tissues. In fact, besides the mechanical action, it is now proposed that potential endocrine and/or paracrine crosstalk exists between bones and muscle (*Brotto & Bonewald, 2015*; *Grygiel-Gorniak & Puszczewicz, 2017*). Studies in humans have found that myostatin a factor secreted by skeletal muscles can regulate bone formation (*Bialek et al., 2014*). Osteocalcin a factor produced by osteocytes was found to have effects on muscle (*Levinger et al., 2014*). Moreover, during embryonic development, osteoblasts from different skeletal regions are derived from different germ layers. Osteoblasts of the craniofacial bones are derived from neural crest cells differentiated from the neuroectoderm. The appendicular bones are derived from the paravertebral mesoderm and the lateral mesoderm, respectively. Cranial bone cells are different from that of the other parts of the body, maturing under different microenvironments and regulatory factors (*Vatsa et al., 2008*; *Wu et al., 2018*). It is somewhat surprising to find that skull BMD was negatively associated with ASM mass, while BMDs throughout the rest of the body were positively correlated with ASM mass among the middle-aged and elderly Chinese men. *Xu et al. (2018)* reported that every part BMD except the head in sarcopenia group were all reduced, which is consistent with our result. Therefore, we speculate that the interrelationship of skull and ASM may be different from that of the bone tissues

throughout the rest of the body and ASM through the circulatory system. However, these hypothesis require further experimental confirmation.

The present study showed that the BMDs of the other parts (including pelvis, hip and femur neck, whole body, thoracic spinal, lumbar spinal, left leg, right leg, left arm and right arm), except ribs, were positively associated with lean mass, which was consistent with most previous studies (*He et al., 2016*; *Verschueren et al., 2013*). However, studies also report that BMD is not associated with lean mass (*Coin et al., 2008*; *Walsh, Hunter & Livingstone, 2006*; *Wu et al., 2002*). *Walsh, Hunter & Livingstone (2006)* reported that in women the relationship between ASMI and BMD disappeared after adjusting for physical activity. *Taaffe et al. (2001)*. have suggested that the positive relationship between lean mass and BMDs might disappear when bone or body size is adjusted for. However, in our analysis, the relationship between BMD and lean mass persisted after adjusting for weight and physical activity. There are several possible explanations for such inconsistent findings. (1) Some studies measured muscle mass as "fat-free mass" which included bone, or as "fat-free soft-tissue mass" which included organ mass. "Fat-free mass" incorrectly strengthens the relationship, while "fat-free-soft-tissue mass" falsely attenuates this relationship (*Baumgartner et al., 1996*). However, in our study, lean mass measured by DXA did not include bone mineral or organ mass but only lean mass of limbs. (2) Differences in experimental design, sample size, demographic characteristics and menstrual status may lead to inconsistent or contradictory results. (3) Most studies did not adjust for the effects of blood sugar, blood lipids and other biochemical markers, which will lead to β value different in regression analysis (*Scott et al., 2017*). In this study, almost all clinical biochemical and metabolic markers were included, and different combinations of variables were adjusted to ensure the reliability of the results. At present, many studies have not strictly excluded metabolic diseases, such as diabetes (most previous studies used questionnaires to exclude diabetic patients instead of using biochemical markers, which will lead to missed diagnosis) (*Scott et al., 2017*). Only by adjusting for such factors can we analyze the independent effect of BMD and muscle mass. (4) The diagnostic indicators and cut-off values of low lean mass differed across studies (*Kim, Jang & Lim, 2016*). In this study, we applied three indicators respectively to define low lean mass and analyze the data as continuous variables, rather than categorized variables, which can avoid the unreliability of the analysis results due to the use of different cut-off values. We found that using SMI was more easily to find the patients with low lean mass than ASMI and ASM/BMI. But the body compositions and BMDs are different among the three sarcopenic groups defined by the three indicators, indicating that the three indicators cannot be used interchangeably. (5) Different BMD sites were observed in different studies.

Several limitations should be noted in our study. First, this study lacks data on grip strength and walking speed. However, since all of our subjects live independently, perform mechanical exercises, they can be considered as presarcopenia. Second, there was substantial loss to follow-up, which may have contributed to a lack of sufficient statistical power. Nevertheless, sensitivity analyses were used to determine that the percentage of missing follow-up data did not affect the results. We analyzed the subjects who performed the examination each year (Tables S5–S7). Third, differences may exist between males and

females for associations of low lean mass and bone, and thus, we are performing prospective studies of elderly women. Fourth, although DXA is an accepted technique for assessing body composition and BMD, its assumptions may influence the interpretation of results. Thus, future studies using imaging techniques such as magnetic resonance imaging and computed tomography may shed additional light on the associations between lean mass and BMDs. Finally, this study lacks the information on vitamin D status during experimental design. The strengths of this study include its long-term follow-up with DXA, large sample size, definition of lean mass using three indicators, and the use of broad-spectrum biochemical and metabolic markers.

## CONCLUSIONS

In summary, the present study found that compared to ASMI and ASM/BMI, SMI was more sensitive to screen for the low lean mass. Skull BMD was negatively associated with ASM mass, while BMDs throughout the rest of the body (including whole body mean BMD, thoracic spinal BMD, lumbar spinal BMD, hip BMD, femoral neck BMD, pelvic BMD, left arm BMD, right arm BMD, left leg BMD and right leg BMD, except ribs) were positively correlated with ASM mass among the middle-aged and elderly Chinese men. Further studies are needed to reveal the potential endocrine and/or paracrine crosstalk exists between bone and muscle.

### Funding

This work was supported by the National Natural Science Foundation of China (81770878), the Guangzhou Planed Project of Science, Technology (201604030007), the National Institutes of Health (R01AR057049, R01AR059781, R01MH107354, R01MH104680, R01GM109068, U19AG055373, R01AR069055), the Special fund of Summit plan [2019A010], the Guangdong Basic and Applied Basic Research Fund [2019A1515110463] and the Special fund of Summit plan [2019C011]. The funders had no role in study design, data collection and analysis, decision to publish, or preparation of the manuscript.

### Grant Disclosures

The following grant information was disclosed by the authors:
National Natural Science Foundation of China: 81770878.
Guangzhou Planed Project of Science, Technology: 201604030007.
National Institutes of Health: R01AR057049, R01AR059781, R01MH107354, R01MH104680, R01GM109068, U19AG055373, R01AR069055.
Special fund of Summit plan: 2019A010.
Guangdong Basic and Applied Basic Research Fund: 2019A1515110463.
Special fund of Summit plan: 2019C011.

### Competing Interests

The authors declare there are no competing interests.

## Author Contributions

- Xuejuan Xu conceived and designed the experiments, performed the experiments, authored or reviewed drafts of the paper, and approved the final draft.
- Nuo Xu, Ying Wang, Jinsong Chen and Jingxian Chen analyzed the data, prepared figures and/or tables, and approved the final draft.
- Lushi Chen and Shengjian Zhang performed the experiments, prepared figures and/or tables, and approved the final draft.
- Hongwen Deng, Xiaojun Luan and Jie Shen conceived and designed the experiments, authored or reviewed drafts of the paper, and approved the final draft.

## Human Ethics

The following information was supplied relating to ethical approvals (i.e., approving body and any reference numbers):

The Foshan First People's Hospital granted Ethical approval to carry out the study within its facilities.

## Data Availability

The raw measurements are available in the Supplementary Files.

## Supplemental Information

Supplemental information for this article can be found online at http://dx.doi.org/10.7717/peerj.10753#supplemental-information.

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
