# Peer review of "The longitudinal associations between bone mineral density and appendicular skeletal muscle mass in Chinese community-dwelling middle aged and elderly men"

_PeerJ, doi:10.7717/peerj.10753_

## Round 0.1 · original submission · Major Revisions

I found the article interesting, which deals with a very hot topic and is therefore worth publishing. However, there are several comments on how to improve and make the manuscript clearer.

There is a great concern about the design of the study that does not allow such a strong conclusion made by the authors. In addition, the three reviewers have identified important points that need to be discussed and answered. The clinical significance and hypotheses of the manuscript are not clear and these need to be clearly stated in the manuscript.

Reviewer 1 ·

Basic reporting

The manuscript “Skull negatively associated with appendicular skeletal muscle mass in Chinese community-dwelling middle aged and elderly men: a longitudinal study ” by Xuejuan Xu et al aimed to investigate longitudinal associations between bone mineral densities (BMDs) and appendicular skeletal muscle (ASM) mass in different regions of the body, in Chinese elderly men. The conclusion was that skull BMD was negatively associated with ASM mass, while BMDs throughout the rest of the body were positively correlated with ASM.
The topic may be interesting but the study suffers from several flaws that limit the significance of the conclusions
The article is well written in English.
The structure ofthe article seems to be adequate

Experimental design

Comments.
1). The lack of any information on vitamin D status may represent a limitation.
2).The statement “.osteoblasts of the craniofacial bones, which are different from other bone tissues of the body,.... and are regulated by different mechanism” does not seem to be supported by adequate references (In fact the study by Vatsa A was carried out in a murine model and the study by Wu V evaluated only the morphology of the osteocytes of the jaw).
3). Did the Authors evaluate the coefficients of variation for skull BMD? The precision of Head BMD should be evaluated.
4). Statistical Analysis: i)The Authors should report whether the study variables were normally distributed; if not the statistical analysis should be changed accordingly. ii)The fact that some independent variables are probably higly correlated may lead to the occurrence of misleading findings; therefore, the multicollinearity of predictor variables should be assessed
5). “Results” section: The values of Head BMD must be reported in both Table S2 and Table S3.
6). “Results” section: The values of Head BMD at each time point (Baseline, 1,2 and 3 year) should be reported in a separate Table.

Validity of the findings

The conclusion was that skull BMD was negatively associated with ASM mass, while BMDs throughout the rest of the body were positively correlated with ASM.
1) The fact that RIB BMD was inversely associated with muscle mass ( see Tables S6 and S7 should be taken into consideration and discussed.
2). The statement “Compared to ASMI and ASM/SMI, SMI was more sensitive to screen for the low lean mass” (“Discussion” Section; Line 4) should be supported by the results of the present study and discussed.
3). The clinical significance of the negative association between Skull BMD and appendicular skeletal muscle mass should be better discussed.

Therefore, the manuscript in the present form does not seem to be suitable for publication in PeerJ.

Reviewer 2 ·

Basic reporting

The present study aimed to investigate longitudinal associations between bone mineral
densities (BMDs) and appendicular skeletal muscle (ASM) mass in different regions of the body, in Chinese community-dwelling middle-aged and elderly men. The authors concluded that the interrelationship of skull and ASM may be different from that of the bone tissues
throughout the rest of the body and ASM.

Major comments
1. Despite Skull BMD negatively associated with appendicular skeletal muscle mass in Chinese community-dwelling middle aged and elderly men: a longitudinal study, the title of this paper does not mention the relationship between skeletal muscle mass and skull BMD in the introduction. The title reminds us of the biological relevance of the relationship between skeletal muscle and skull.
2. The clinical significance and hypothesis in the manuscript is unclear. The results of this study need to describe what social contributions can be expected.
3. There is no discussion about the relation between sarcopenia or skeletal muscle mass and skull BMD.
4. I want you to specify results that are only important results. Skull BMD does not focus in the results.

Experimental design

I think no problem.

Validity of the findings

The clinical significance and hypothesis in the manuscript is unclear.

Additional comments

Need to structure papers focused on skulls.

Reviewer 3 ·

Basic reporting

No comment

Experimental design

No comment

Validity of the findings

No comment

Additional comments

Overall the manuscript is well structured, and it will add to the current evidence on appendicular muscle mass and bone mineral density in adults. However, I have several concerns over the study and my comments are outlined below,
• Abstract – Results: “The incidence of low lean mass was 8.2% defined by ASMI, 16.3% defined by SMI, and 8.3% defined by ASM/SMI.” Is the 8.3% referring to “ASM/BMI” instead of “ASM/SMI”? “ASM/SMI” is not one of the three indicators to define ASM mass or lean mass in the Methods section.
• Abstract – Results: Please include the unit for ASMI, SMI, and ASM/BMI (as appropriate).
• Abstract – Conclusions: “Compared to ASMI and ASM/SMI, SMI was more sensitive to screen for the low lean mass.” Should this read “ASM/BMI”, instead of “ASM/SMI”? Please kindly use the same term throughout the manuscript.
• Lines 88-104: It will be great if the authors can include the explanations for the contradictory findings.
• Lines 157-158: Was this trial registered in any clinical trials registry, e.g. Clinicaltrials.gov, Australian New Zealand Clinical Trials Registry, etc.? If so, please kindly include the information in Methods section.
• Lines 167-177: Is there any reason these biochemical indices were selected?
• Lines 171-172: “Postprandial blood glucose” – what was the test food (e.g. 75 g glucose) and at which time point the blood glucose was measured (e.g. 1-hour, 2-hour, etc.)?
• Line 169-177: Please elaborate on “etc.” and it would be helpful to categorize the biochemical indices, e.g. blood lipid profile, full blood count, etc.?
• Line 180: Please switch the order of “BMD (g/cm2)” and body composition so it’s in line with the heading in Line 179.
• Lines 184-189: Were these intra- or inter-assay CV?
• Lines 192-194: “We used three indicators to diagnose low lean mass respectively: (1) an ASMI < 7.0 kg/m2. This approach for defining low lean mass was based on the new Asian consensus definition of the AWGS (Chen et al., 2016).” The previous statements in the Methods (Lines 127-130) stated that “The AWGS proposed a cut-off value for the definition of low lean mass using the ASMI in Asian populations. Using dual-energy X-ray absorptiometry (DXA), the threshold was 7.0 kg/m2 for men and 5.4 kg/m2 for women (Chen et al., 2014).” Please include gender-specific cut off in the Results section, e.g. 7.0 kg/m2 for men and 5.4 kg/m2 for women, instead of using 7.0 kg/m2 for all study participants. Thank you.
• Line 222: Please define “GGT” and include this in the “Blood sample measurements” section.
• Line 225: Is there any reason HbA1c was adjusted for, and why gender was not included in the model?
• Line 233-235: Please include the reason why these variables (i.e. age, weight, HbA1c, HDL-C, Cr, ALT, FT4, diastolic blood pressure, smoking, drinking, and exercise) were included in the model. Thank you.
• Lines 248-250: “… stratified into two groups based on the ASMI. ASMI <7.0 was used to define low-lean-mass group, and the control group was defined ASMI >=7.0.” Please use gender specific cut-off to define low lean mass group.
• Lines 262-263: Please define epidemiological survey data.
• Line 304: Please change “indcluding” to “including”.
• Lines 362-363: “We found that using SMI was more easily to find the patients with low lean mass than ASMI and ASM/BMI.” Is this statement referring to the higher prevalence of low lean mass using SMI, compared to ASMI and ASM/BMI?
• Table 1: Please use gender-specific cut-off to diagnose low ASMI, instead of “ASMI <7.0” for both genders.
• Table 1: Please kindly include what Q value means either in this table or the Methods section.
• Figure 3: Please include the unit for ASMI, ASM, and ASM/BMI in the y-axis of Figure 3 (a), (b) and (c).

Thank you!

---

## Round 0.2 · Minor Revisions

The authors have done a great job in revising their manuscript, but there are still issues that need to be addressed before we can recommend publication. For example, at the beginning of their rebuttal letter they indicate that the last paragraph includes information regarding the limitation of the lack of data on vitamin D levels. There is no mention of this text in the tracked changes version, neither, for example, is there any mention of the book Developmental Biology which is used to justify some of the statements included in the text. I therefore ask that the authors carefully review all those issues that should be indicated in the manuscript have been.

---

## Round 0.3 · accepted · Accept

The authors have done an appreciable effort in reviewing their manuscript. I consider that in its current version it is likely to be recommended for publication.